# How grid reinforcement costs differ by the income of electric vehicle users

**Sarah A. Steinbach** [1] ✉ **& Maximilian J. Blaschke** [1,2]

The simultaneous charging of many electric vehicles in future mobility scenarios may lead to peaks and overloads threatening grid stability. The necessary infrastructure investments vary by the number and model type of vehicles driven and the residents' charging preferences. These attributes significantly depend on socio-economic factors such as income. Using power flow simulations based on real-life driving profiles, we predict massive cost asymmetries with an investment demand up to 33-fold in higher-income compared to lower-income neighborhoods. Many grid operators may redistribute these costs through an across-the-board electricity price increase for all households. In times of rising electricity prices, these unwanted inequitable costing allocations could lead to severe challenges and energy poverty. Policymakers should consider countermeasures like dynamic electricity pricing schemes, income-based electric vehicle subsidies, or improved charging network access to ensure energy equity in future mobility scenarios. Our analysis of the impact of socio-economic factors on electric vehicle grid infrastructure and their quantification contributes to the energy equity discussion.

With tightening carbon emission regulations in the transportation sector, more and more consumers are switching to electric vehicles (EVs). However, charging a high number of EVs poses challenges to the distribution grids: Most consumers prefer charging their EVs at similar times during the day, especially in the early evening hours. Simultaneous charging of multiple EVs could lead to significant load peaks, causing overloads within the grids[1–3]. These overloads increase with EV adoption and depend on the EV model choice as well as the applied charging patterns. All these factors may be correlated with socio-economic attributes like household income[4–7]. Therefore, grid operators may have to over-proportionally enhance the grid infrastructure in areas with many above-average (higher-) income households. These costs must be borne, most likely via a surcharge on the electricity price for all households. Hence, the high infrastructure investments mainly caused by higher-income neighborhoods would also place a particular burden on below-average (lower-) income households. This cost causes an imbalance in combination with current cost allocations within the electricity tariffs and could, therefore, lead to an unfair distribution of costs. Households within the lower-income classes may have to

reduce or even stop basic activities like cooking or washing or have to compensate through reducing other expenses. In times of rapidly increasing energy prices, energy equity becomes a topic of increasing social and public interest, as energy poverty begins to affect even middle-class households with a household income of 60-80% of the median household income[8]. The clean energy transition might disadvantage lower-income households[9]. Lower-income neighborhoods experience stronger grid limitations, reducing their access to residential photovoltaics and potentially hindering EV adoption[10]. This could disadvantage entire population groups. Hence, efforts to accurately measure energy inequity and strive for energy justice through policy measures are increasing[11,12]. With a rapid transition towards EVs ahead of us, it is now most relevant to be aware of the social consequences of the corresponding infrastructure investments. It is still unclear to what extent these investments may lead to inequities and how these cost imbalances should be dealt with from a political perspective. Our paper investigates and quantifies the difference in the necessary grid reinforcement costs between lower and higher-income neighborhoods, meaning a 50:50 split of households by income. From

[1]Chair of Management Accounting, TUM School of Management, Munich, Germany. [2]Center for Energy and Environmental Policy Research, Massachusetts Institute of Technology, Cambridge, MA, USA. ✉e-mail: sarah.steinbach@tum.de

these calculations, we determine the over-proportional grid reinforcement costs of higher-income EV users and the potential for energy inequities. Based on our findings, we derive policy recommendations to help prevent financial pressure on lower-income households and mitigate future energy inequity.

The high infrastructure costs are rooted in the problem that EVs require reinforcements in grid infrastructure[1,2,13,14]. Researchers uncover that plug-in hybrid electric vehicle penetration levels between 10% and 30% lead to significant voltage imbalances and power losses[1]. Building on these findings, numerous authors find similar results with grids being unable to handle EV charging loads in different countries and grid scenarios[2,3,14,15]. Further, general overviews and outlooks of the challenges associated with integrating EVs into the grid have been published[13,16,17]. The recent technological shift towards sole battery-powered electric vehicles (BEVs) and higher charging powers could further increase the pressure on the grid, requiring new solutions and improvements within the infrastructure[18–20].

When analyzing EVs' impact on the distribution grids, most studies model all households within the simulated distribution grid with homogeneous EV adoption and usage behavior. However, socio-economic factors such as income, age, gender, occupation, level of education, ethnicity, home ownership, and residence type, as well as political orientation, play a role in mobility per se[21,22], in the adoption of new technologies[23] and specifically EV adoption[4,5,24–30]. The homogeneous modeling of grid impact could hence be prone to significant errors[7]. Of all possible socio-economic factors, income is still the primary driver of EV adoption[4]. Above-average household income increases the likelihood of owning an EV by as much as 200%[24], and overall medium to high-income groups tend to show higher EV adoption across regions[4,25,26]. Analyzing survey data from more than 5000 respondents in the Nordics, previous literature uncovers that higher income is associated with an increased likelihood of owning an EV and more expensive car models[5]. Researchers find similar results in other regions[27,31,32]. These more expensive and frequently larger car models tend to have higher electricity consumption, increasing charging loads and pressure on the grid[33].

Besides EV adoption and car model choice, driving patterns greatly affect EV charging patterns and potential load peaks. The driving patterns depend on socio-economic factors, including age, gender, and level of education or occupation. Depending on these factors, the number of trips per day, as well as the departure and arrival times impacting charging times, vary significantly[6,34–36]. Since socio-economic factors may lead to higher worst-case power flows, researchers criticize current charging modeling approaches and call to include these factors in load assessments[7,37]: One study simulates EV charging demand accounting for socio-economic factors such as household income or occupation and analyzes the related load curves in a German setting[36]. A recent 2035 forecast for the US also criticizes current charging modeling approaches[37]. Using a data-driven model distinguishing driver income, housing, and miles traveled, they find that EV charging loads increase peak net electricity demand by up to 25% and deduct related implications as, for example, the charging point dissemination. A study simulating EV charging loads of UK households with differing economic statuses, finds that higher-income households cause larger load peaks, potentially leading to over-proportionally high grid reinforcement costs[4]. Their paper hence raises the issue of a fair grid cost allocation. While only a few studies include socio-economic factors in their load assessment, none of these studies provide estimations on the related distribution grid reinforcement needs or related grid reinforcement costs.

Fairness in the allocation of these grid reinforcement costs is a matter of perspective. The allocation of costs can be distinguished between three principles: The allocation of costs along the need, the contribution to a problem, or to a simple equal share[38]. In the context of reducing $CO_2$ emissions, most individuals prefer the principle of

contribution (equity), where people who contribute more emissions should have to achieve higher emission reductions[39]. The issue of fairness in bearing the here-mentioned grid infrastructure costs is slightly more complicated, as higher-income households with more EVs might cause over-proportionally high infrastructure costs but also reduce $CO_2$ emissions. However, in the coming years, EVs are expected to become cheaper with economies of scale. In the long run, we may assume an equally high share of vehicle electrification in lower- and higher-income households. At that point, we may still face differing costs in required grid reinforcements due to driving behavior and vehicle ownership. Applying the principle of fairness according to contribution would require higher-income households to carry the caused asymmetry in grid reinforcement costs to the full extent.

However, residential grid reinforcement costs in many countries are compensated for as part of the electricity price via a fixed component as well as a fee per kWh (generally[40,41] and, for example, in Germany[42,43]). Without any political corrections, increased grid reinforcement costs would lead to an overall electricity price increase for all consumers. This price increase could be considered inequitable to the principle of fairness according to the contribution, as higher-income neighborhoods over-proportionally cause these grid reinforcement costs. Previous literature does not sufficiently consider socio-economic factors for the related grid cost scenarios and, hence, could not quantify the potential inequities. Tackling this research gap, we focus our analysis on household income as a critical socio-economic factor and raise a discussion on energy equity.

In this work, we quantify the over-proportional grid reinforcement cost impact of higher-income EV users. We, therefore, use real trip data[44] in a grid power flow analysis to compare the grid reinforcement costs of above-average with below-average income neighborhoods. Our paper builds on previous literature on electric vehicle charging and grid resilience and contributes to the increasingly relevant field of energy equity. The context of socio-economic factors, such as income, should be highly relevant for policymakers, who increasingly incorporate social aspects as critical factors in electricity pricing and regulatory measures and policy instruments overall[45–48]. Furthermore, our paper illustrates the need for grid planners to include socio-economic factors such as income in their grid planning models, as some providers already started to do so[49].

## Results
### Model overview
We simulate electricity usage for two neighborhood types: below-average (lower) and above-average (higher) income, meaning a 50:50 split of households by income. For these two neighborhood types, we assign respective EVs considering adoption and model choices and fit the corresponding mobility behavior based on representative real-life driving data. We use representative distribution grids in urban, suburban, and rural settings to account for the differing structure and load capacity[50,51]. After allocating empirically sampled EVs amongst the grid nodes, our power flow simulations check each setting for overloads. While we showcase the approach with inputs for distribution grids in Bavaria in the South of Germany, the approach may be applied to any grid or geographical region. The differences observed between above- and below-average neighborhoods with a 50:50 split already showed such significant cost imbalances that a further split of the population into smaller groups would most likely only lead to even more extreme results. Due to smaller samples and less data availability, we, however, would not receive stronger or more robust findings. The simulation builds upon the approach in our recent work[52] and is structured as displayed in Fig. 1.

We derive the grid reinforcement cost asymmetry between the two neighborhood types within a power flow analysis performed using the Newton-Raphson method of the matpower package in MATLAB, which is frequently used in load analysis[53]. To consider the most

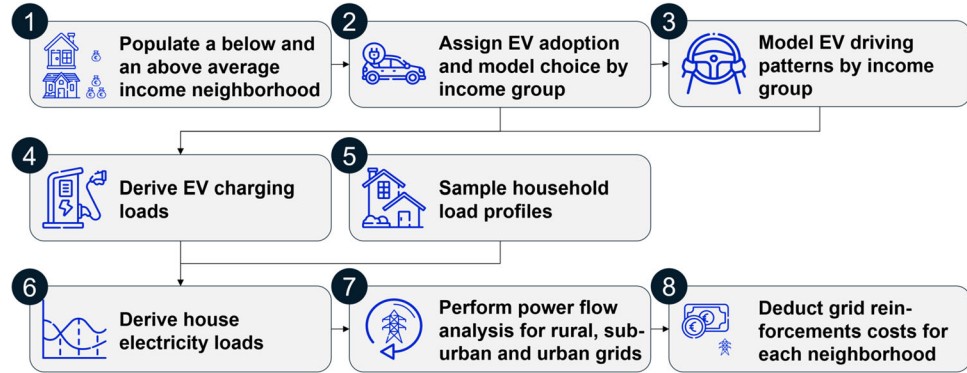

**Fig. 1 | Simulation approach to quantify the costs of reinforcing distribution grids.** This flowchart illustrates the elements of our simulation model approach labeled by numbers in the order of execution as described in the following. For both neighborhood types, we first populate the grids with the related income group (above- or below-average-income households) by sampling from the household distributions of each neighborhood and grid type. We then assign the related electric vehicle (EV) adoption level and model choice by randomly allocating EVs depending on the income group's EV adoption and model choice distribution. As a third step, we model the driving patterns for each EV depending on

the income group by a time-inhomogeneous Markov chain simulation. Next, we derive the EV charging loads resulting from the EV driving patterns based on the charging probability as well as future car trips planned. We empirically sample representative household electricity load profiles on a household level. We consolidate the EV charging loads and household electricity load profiles by adding up all loads at the house level. Via a power flow analysis, we analyze if overloads occur and reinforce the respective overloading grid element. Lastly, we calculate grid reinforcement costs for each grid and neighborhood type to resolve these overloads.

challenging season for electricity usage, we perform the simulation using 5-min intervals for the month of December and share the results for an average week. A more detailed description of our Monte Carlo approach, the uncertainty parameters in our simulation, and the data used is outlined in the "Methods" section.

### EV adoption scenario analyzed

In order to simulate realistic future EV penetration levels, we use the current German government target of 15 million EVs on German roads by 2030 as the basis for our scenarios[19]. Relative to the 2021 German car park of 48.24 million cars, this would equate to an EV adoption rate of 31.1%[54]. Analysis of current EV sales reveals that a household's probability of owning an EV is up to three times as high for higher-income than for lower-income households[24]. Using the 15 million EV target as a base (equating to an overall EV adoption rate of 31.1%) and accounting for higher-income households owning more cars, EV adoption rates would lie at 22.4% for lower-income and 35.7% for higher-income households. As we could expect this effect to become smaller as more EVs enter the market and prices decrease, we also included an analysis of equal EV adoption rates for both income groups.

### Driving patterns and load profile implications

First, we investigate the differences in driving patterns, EV charging behavior, and resulting load curves. In line with existing literature, we find that higher and lower-income households differ in their driving patterns. Our car trip dataset reveals that higher-income households perform more daily trips, with an average of 2.2 daily trips instead of 2.0 daily trips for lower-income households[44]. They also exhibit longer trip durations of, on average, 42 min instead of 38 min. Furthermore, higher-income households show more concentrated weekday home arrival times, leading to stronger load peaks, as visible in Fig. 2.

These effects are most likely also linked to differences in occupation and level of education between the two income groups, which have been shown to affect mobility behavior[34–36]. While 46% of drivers in higher-income households are working full-time, this only applies to 25% of drivers in lower-income households within the data set[44]. The proportion of drivers with a university degree, which is often linked to a nine-to-five work schedule, is 43% for higher-income and only 26% for lower-income households. This may be the reason why we observe more concentrated weekday arrival times and increased car usage for

high-income households. Due to the longer driving times, the hence higher electricity consumption, and the more pronounced weekday arrival time peaks, we expect the high-income households to induce stronger load peeks, especially on weekdays. This effect is also enhanced by the difference in EV adoption as well as model choice. Figure 3 shows the exemplary case of the induced load curves for an average week in December in the rural grid.

As expected, the load peak difference between higher- and lower-income households is especially pronounced on weekdays. Due to the stronger load peaks, we expect the higher-income neighborhood grids to be more at risk for overloads caused by EV charging.

### Impact on grid overloads

Based on these simulated load profiles, we investigate the overloads occurring for higher and lower-income rural, suburban, and urban neighborhoods. This overload analysis is relevant for grid planning, as it displays which neighborhoods require prioritization. Figure 4 illustrates the number of 5-min intervals in which an overload occurs. For example, on average, the rural grid experiences 5 overloads during a week in the lower-income neighborhood, while 70 overloads occur in the higher-income neighborhood.

In all area types, higher-income neighborhoods would experience significantly more grid overloads, putting these neighborhoods higher on the grid operators' agenda for grid reinforcements. As the number of overloads and, hence, the probability for a blackout differ significantly between lower and higher-income neighborhoods, the importance of including socio-economic factors such as income in grid planning models becomes apparent. The rural grid is the weakest and exhibits the most overloads. However, a transformer replacement in this grid would solve the vast majority of overloads occurring, while grid lines are the most prevalent bottleneck in the other grid types.

### Asymmetry in grid reinforcement costs and underlying effects

In this section, we derive the related grid reinforcement costs to mitigate the overloads previously outlined and to stabilize the grid. The average reinforcement costs to be expected are illustrated in Fig. 5. While the analysis does not prove a causal relationship between household income and grid reinforcement costs, it provides illustrating scenarios for future grid reinforcement costs to be expected for a representative higher-income compared to a lower-income neighborhood.

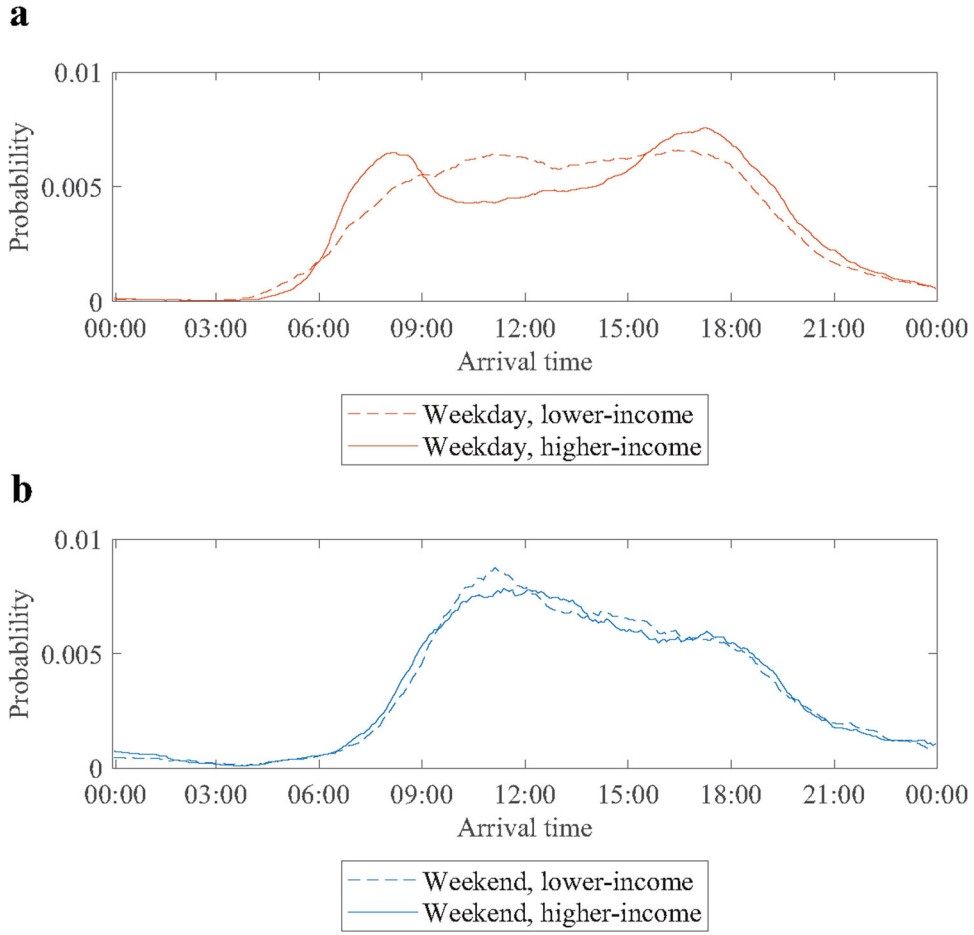

**Fig. 2 | Probability of car arrival at home for an average weekday and weekend.** **a** Probability of a car arrival at home for an average weekday. The dashed line shows the home arrival probability for lower-income drivers throughout an average weekday. The solid line is associated with higher-income drivers. **b** Probability of a car arrival at home for an average weekend. The dashed line reflects the home arrival probability for lower-income drivers throughout an average weekend day. The solid line shows this probability for higher-income drivers. Source data are provided as a Source Data file.

We see 50% additional grid reinforcement costs for higher-income neighborhoods in the rural, 3266% in the suburban, and 478% in the urban grid compared to lower-income neighborhoods. The additional reinforcement costs are the lowest for the rural grid as this grid is the least resilient overall. An upgrade of its bottleneck, the transformer, becomes inevitable even for lower EV charging loads. These significant asymmetries in grid reinforcement cost further illustrate the necessity for grid operators to include socio-economic factors such as income in their grid planning models to represent future grid costs adequately. These significant asymmetries also prevail when testing for the inclusion of residential electricity generation and storage. When extrapolating our findings to the around 119 million residential buildings in the EU and accounting for their distribution to rural, suburban, and urban areas, the potential grid cost asymmetry between higher- and lower-income neighborhoods could reach around €14 billion[55–58].

In order to derive appropriate mitigating policy measures, we further analyze the impact of the underlying drivers for the additional grid reinforcement cost of higher-income neighborhoods. We quantify the standalone impact of differences in EV adoption, model choice, and driving patterns by neighborhood type. For that purpose, we keep all other parameters equal (ceteris paribus) and adjust one driver as follows: EV adoption - we derive the effect of EV adoption by assigning both income groups the same EV adoption rate of 31.1%. Model choice— we quantify the impact of model choice by assigning the car segment distribution of lower-income households to higher-income households.

Driving patterns—we analyze the impact of driving patterns by assigning the driving patterns of lower-income groups to higher-income groups. We then compare these results to our original values.

It is important to note that these three drivers are not additive. However, this analysis provides an understanding of the most effective levers for diminishing grid cost asymmetry and related inequities. In Fig. 6, we analyze the effect of EV adoption.

If EV adoption were equally distributed over all neighborhoods, the grid reinforcement cost asymmetries would shrink significantly. This effect, however, is partly caused by a related grid cost increase for lower-income neighborhoods. Nonetheless, our results show that even if equal EV adoption levels across income levels could be achieved, significant additional grid reinforcement costs for higher-income neighborhoods prevail, especially for the suburban and urban grids.

Figure 7 illustrates the impact of model choice and driving patterns of higher-income households. We discuss only the urban grid, as the effects for the other two grid types are similar.

Driving patterns strongly impact grid cost asymmetry, while the effect of model choice is relatively small. This can also be observed for the rural and suburban grids, with additional costs shrinking in the suburban and slightly also in the rural grid. For more details on this matter, please refer to the Supplementary Fig. 1. These findings indicate that policymakers may foster EV adoption with all model sizes but focus more on reducing peak-hour charging to mitigate some behavioral effects of higher-income households.

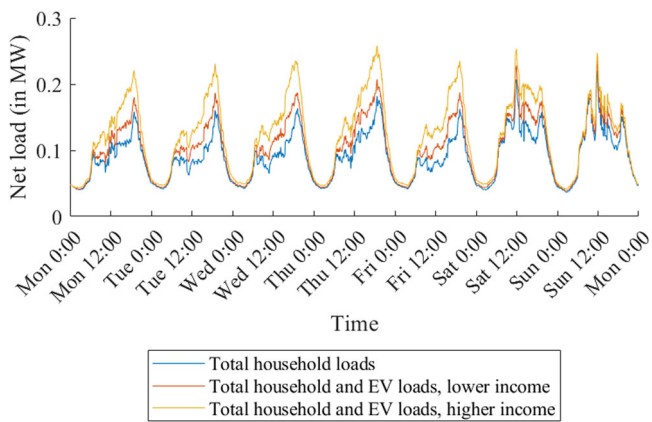

**Fig. 3 | Net load profiles of households with and without electric vehicles (EVs) in the rural grid.** The lines indicate the induced load curves for an average week in December in the rural grid. While the blue line shows the total household loads before EV charging, the orange line also includes the EV charging loads for the lower-income neighborhood. The yellow line includes the household loads with EV charging loads for the higher-income neighborhood. Source data are provided as a Source Data file.

## Electricity pricing implications and potential inequity

Residential grid reinforcement costs are currently paid for via the consumer electricity price, which is determined per kWh[42,43]. These prices do not vary with the load or maximum power demand but are reimbursed with a flat-rate cost allocation[42]. As can be seen in Fig. 8, the proportion of the electricity price allocated to grid costs for an average household in 2021 was around 23%[43].

If grid costs increase, the electricity price for all consumers is inflated, and electricity costs increase for all households. Due to their higher total electricity consumption and related higher electricity costs, higher-income neighborhoods carry more of the grid reinforcement costs in total. However, as they only consume 16–18% (based on the area type) more electricity than lower-income households, this contribution fails to offset the massive additional grid reinforcement costs caused. Furthermore, grid operators often split grid costs into a base rate in addition to a volumetric (per kWh) component. This base rate is not scaled with regards to consumption and hence further limits the grid cost contribution of higher-income households[42]. With household electricity prices at a record high (32.63 cent/kWh in 2021 and quickly increasing during the European Energy Crisis in 2022[59–61]), a further across-the-board electricity price increase to cover the additional grid reinforcement cost of higher-income neighborhoods could be considered inequitable with respect to the principle of

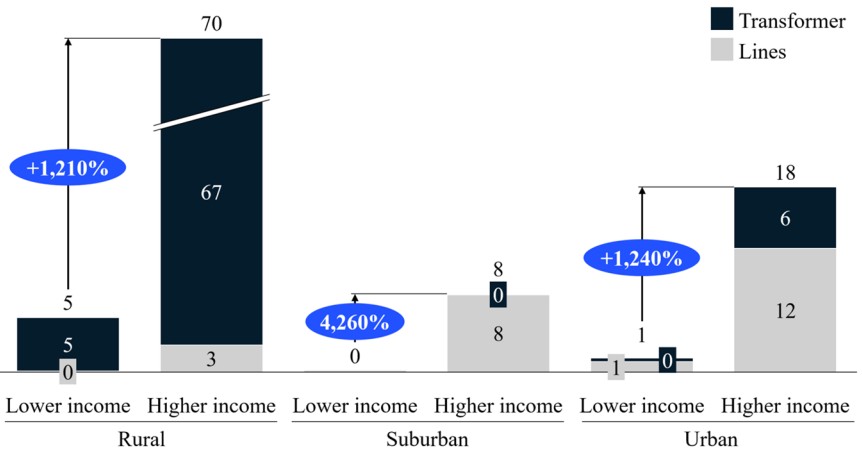

**Fig. 4 | Average number of weekly overloads in December.** For the rural, suburban, and urban grids, we compare the average number of overloads occurring during a week for lower-income and higher-income neighborhoods. Overloads within the transformer are displayed in black, and overloads within the lines in gray. The relative difference between the number of overloads in the lower- and higher-income neighborhoods is given in blue. Source data are provided as a Source Data file.

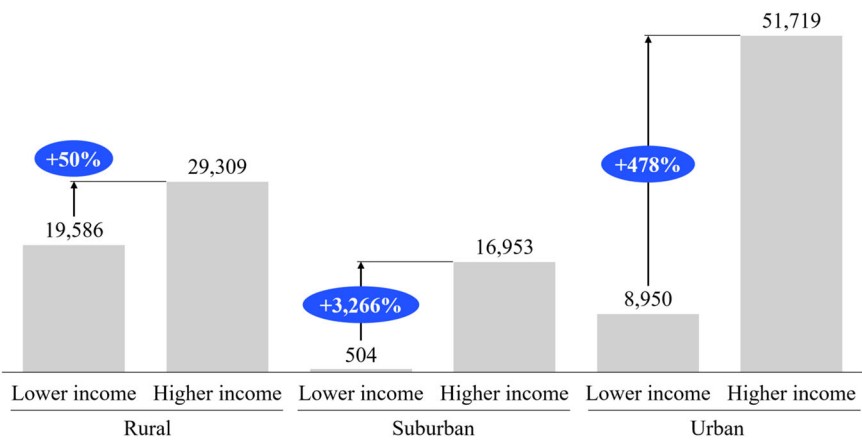

**Fig. 5 | Average simulated grid reinforcement costs.** For all three grid types, we compare the average grid reinforcement costs occurring for lower-income and higher-income neighborhoods. Costs in € are displayed as gray bars. The relative difference in cost between the lower- and higher-income neighborhoods is indicated in blue. Source data are provided as a Source Data file.

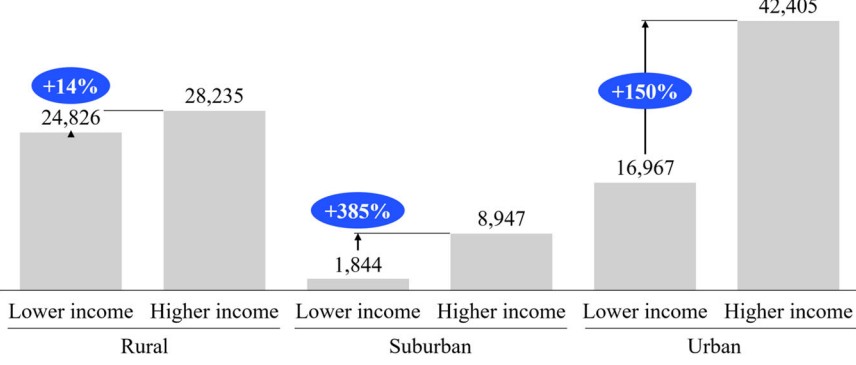

**Fig. 6 | Average simulated grid reinforcement costs assuming equal EV adoption levels.** For each grid, we compare the average grid reinforcement costs occurring for lower-income and higher-income neighborhoods if they would both have the same EV adoption rate. Simulated reinforcement costs in € are displayed as gray bars. The relative difference in cost between the lower- and higher-income neighborhoods is given in blue. Source data are provided as a Source Data file.

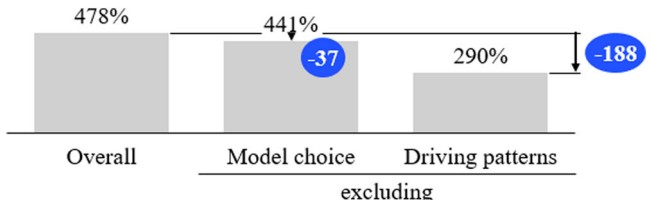

**Fig. 7 | Breakdown of grid reinforcement costs asymmetries.** We break down the difference in grid reinforcement costs between lower- and higher-income neighborhoods by the underlying drivers of model choice and driving patterns. Here, we use the urban grid as an example. The gray bars show the persisting cost difference. In blue, we show the respective effect of each factor. Source data are provided as a Source Data file.

fairness according to contribution. As this grid reinforcement cost asymmetry can be traced back to higher-income neighborhoods, equitable cost allocation would require higher-income households to fully bear this cost asymmetry, not affecting the electricity prices of other consumers. This effect could be aggravated further if grid operators decide to prioritize higher-income neighborhoods in their grid planning, potentially making some households pay higher electricity prices long before they can profit from more stable grids in their neighborhoods. Furthermore, this prioritization could lead to energy access equity issues between neighborhoods, as, for example, already observed in the case of solar photovoltaic systems in the United States[10].

However, it is important to note that the rationale for the mentioned potential inequitable cost allocation is not the difference in income between the two neighborhood types but the difference in usage of the electric distribution grid as a common public resource. According to the principle of fairness according to contribution (equity)[38], households in a higher-income neighborhood should rather pay grid fees, which reflect the contribution to the grid reinforcement costs induced by them. We focus on mitigating policy actions directly related to households' grid cost impact, not socio-economic attributes such as income.

## Possible mitigating policy measures

Regulators could use various policy instruments to tackle potential inequities. They may adjust the electricity tariff design, e.g., with a time-of-use tariff, to mitigate the grid load effects of EV charging, which we showed to be more pronounced for higher-income neighborhoods. Another option would be to distinguish and allocate costs in smaller tariff zones. Hereby, the high investment costs of wealthier neighborhoods would not necessarily bother other neighborhoods. However, such individualized cost allocation would come with massive

effort and complexity. Last, policymakers may address the asymmetry in EV adoption and model choice between income groups. For any policy measure implemented, it is essential to ensure that although EV charging poses a challenge to distribution grids, EV ownership and usage over combustion engine cars should never be discouraged.

In a time-of-use tariff, household electricity prices increase significantly during peak load times. The increased pricing during these times allows the allocation of over-proportionally more costs to the drivers of EVs and keeps the prices low for activities outside the peak hours. Such a tariff would, furthermore, reduce the overall infrastructure costs by incentivizing EV charging outside peak hours, reducing simultaneous charging. A large grid operator in Denmark, for example, already employs this tariff policy, with grid tariffs more than doubling between 5 and 8 p.m. during the winter months[62]. Recent works recommend similar electricity tariff adjustments to promote energy equity[45]. This time-of-use tariff approach is easier for consumers to respond to than more dynamic electricity tariff strategies such as real-time pricing or critical peak pricing. This allows for more equitable electricity pricing across consumer groups of differing knowledge levels. Policymakers should further consider alternative electricity tariff models that adjust for maximum electricity loads induced, also called demand charge rate. This demand charge rate is already used for industrial electricity consumers and used to price their grid load impact[42]. When incorporating a demand charge rate, however, its pricing needs to be carefully calibrated to not discourage EV adoption and usage[45]. Any dynamic electricity pricing, load-based or adjusted for time, does, however, require the installation of a smart meter. The smart meter installations are, unfortunately, lagging behind. In Germany, for example, only 19% of households own any smart energy management device in 2022. Since energy companies fall behind their smart meter installation ambitions[63,64], alternative measures are worth considering.

EV adoption greatly impacts the magnitude of the inequitable grid cost allocation. As it is not desirable to reduce overall EV adoption and limit the electrification of mobility, policymakers could reduce the inequity in cost allocation by increasing EV-related subsidies for lower-income households, where EV subsidies have shown the strongest impact[65]. A fuel efficiency-dependent reduction in government EV subsidies based on car models could also compensate lower-income households and mitigate some inequities. However, the effect of model choice on grid costs is limited, as seen in Fig. 7. Furthermore, policymakers should promote the expansion of charging stations in multi-family buildings as well as public chargers, as insufficient access to charging stations tends to limit EV adoption, especially for lower-income households[24,45]. Unfortunately, households in the lowest income classes that can not afford an electric vehicle anyway will not profit from any of such actions but will still face higher grid costs.

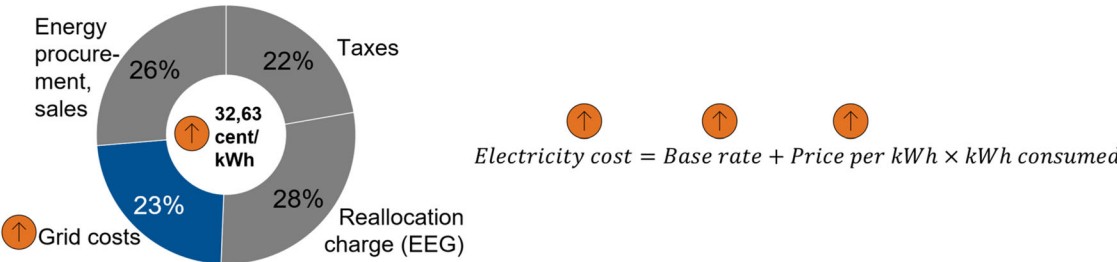

$$Electricity\ cost = Base\ rate + Price\ per\ kWh \times kWh\ consumed$$

**Fig. 8 | Electricity price split and cost calculation, Germany 2021.** On the left, we illustrate the breakdown of electricity cost in its components[43]. On the right, we sketch the most common billing function of current household electricity price tariffs. The orange arrows show how a change in grid costs would drive up household electricity costs.

## Discussion

Our work analyzes the difference in grid reinforcement costs induced by EV charging in lower- compared to higher-income neighborhoods. In the analyzed scenario, the number of grid overloads occurring for higher-income neighborhoods exceeds those for lower-income neighborhoods by over 12-fold on average across different grid types. Hence, the stronger need for grid reinforcements puts higher-income neighborhoods at the top of grid operators' agendas, potentially limiting future charging network access in lower-income neighborhoods. While grid reinforcement costs from higher-income neighborhoods in rural grids are only 50% higher, we see a more significant effect in suburban and urban grids, with costs diverging by up to around 3300% and 480%, respectively. For the EU, these cost asymmetries could potentially amount to €14 billion. The current policy setting would cover the related grid reinforcement costs via an across-the-board electricity price. This could be considered inequitable regarding the principle of fairness according to the contribution, as these grid reinforcement costs can be over-proportionally traced back to higher-income neighborhoods. Policymakers should hence consider adopting a dynamic electricity tariff such as time-of-use or load-based pricing to prevent assigning these costs to all electricity consumers. As grid cost asymmetries are also strongly driven by EV adoption, policymakers may try to compensate for inequities with income-dependent EV subsidies or promote charging network access for lower-income groups.

Our findings on inequitable EV-related grid cost allocation contribute to the larger field of energy inequity, which has gained importance globally in recent months[8,46–48,66,67]. With energy and electricity prices rapidly increasing due to the Ukraine war, lower-income households in Europe are over-proportionally affected by the rise in energy costs[47,66,67]. Energy poverty is quickly becoming an issue affecting also middle-class households[8]. Targeted, income-adjusted government relief measures could be required to support lower-income households and allow equitable cost allocation[8,47]. Current energy crisis relief measures are frequently still falling short of this goal[12,46,48,68].

## Methods

### EV portfolio and driving patterns by income class

To assign our EV portfolio and driving patterns based on real-life data, we leverage the German Mobility Panel, a renowned 30-year project collecting representative mobility behavior of households[44]. First, we create a set of below- and above-average-income households for a German neighborhood. Using current German household net income data, we find that the average net income lies around 3600€ per month[69,70]. We leverage the household data from the German Mobility Panel to separate this data set by household income[44]. The data set states monthly income with steps of 500€ granularity. We separate households with below- (lower) and above-average (higher) incomes at 3500€ net monthly household income.

We again use household data from the German Mobility Panel to assign EVs to households by determining the number of private cars owned per household based on the area type[44]. When analyzing the average number of cars per household by income group, we can see significant differences, with lower-income households owning, on average, 0.94 and higher-income households 1.77 cars. When including EVs in our model, we choose to use BEVs only as this reflects the markets' direction to reduce all conventional vehicle powertrain technologies[19,20]. We separate the EV into different segments: Mini (Volkswagen e-UP), Small (Renault Zoe Z.E. 40 R110), Compact (Volkswagen ID.3 Pro), Medium-sized (Tesla Model 3 Long Range Dual Motor), SUV (Sports Utility Vehicle, Audi e-tron 55 quattro) and Luxury (Porsche Taycan Turbo S). The different vehicle classes allow consideration of the varying power consumption and the corresponding charging needs. We derive the usable battery capacity from a key database compiling technical specifications for EVs currently on the market[71]. The electricity consumption data is based on real-live driving tests[72,73]. Table 1 lists the car segments' specifications. Since our simulation investigates the demanding December conditions, we utilize climate data[74,75] to fine-tune the electricity consumption considering the ambient temperature. This adjustment is needed as ambient temperature significantly affects the energy efficiency of an EV. Specifically, temperatures between 0 °C and 15 °C decrease vehicle ranges by up to 28% in comparison with driving at moderate temperatures from 15 °C to 25 °C[74,76].

To find the appropriate segment sizes for the German car market, we aggregate the newly registered cars per segment of 2021 as provided by the German federal transport agency (Kraftfahrtbundesamt)[77]. To separate between lower and higher-income households, we leverage the car segmentation included in the German Mobility Panel to reflect model choice differences between income classes[44]. We choose to re-scale the newly registered car segment distribution from the German federal transport agency[77]

**Table 1 | Car segment battery capacity and consumption[71–73]**

| Segment | Mini | Small | Compact | Medium-sized | SUV | Luxury |
|---|---|---|---|---|---|---|
| Usable battery (kWh) | 32 | 41 | 58 | 76 | 87 | 84 |
| Electricity consumption (kW/100 km) | 17.7 | 20.3 | 19.3 | 20.9 | 25.8 | 33.0 |

SUV refers to Sports Utility Vehicle.

**Table 2 | Car segment distribution by income class**

| Household group | Mini | Small | Compact | Medium-sized | SUV | Luxury |
|---|---|---|---|---|---|---|
| Lower-income | 8% | 19% | 21% | 12% | 31% | 8% |
| Higher-income | 6% | 13% | 19% | 17% | 25% | 19% |
| All | 7% | 16% | 20% | 15% | 28% | 14% |

SUV refers to Sports Utility Vehicle.

instead of simply using the 2019 German Mobility Panel's car segment distribution to reflect future car model choice instead of the existing German car park. The resulting impact of income on the car model choice can be seen in Table 2.

We use real-life representative driving data from the German Mobility Panel[44] collected between September 2019 and the beginning of March 2020 to simulate car driving patterns by income class. This data set includes weekly trip data for 70,796 trips covering various modes of transportation, provided with travel times, trip purposes, and timings. These trips are recorded with 1-min accuracy. After selecting only trips performed by car and outlier removal by excluding drivers performing holiday trips or very long journeys (above 200 km, longer than 132 min), we arrive at a data set of 22,803 trips representing common driving patterns. We assume that the first trip of each day always starts at home and the last trip of each day ends at home. We generate synthetic trips for both income groups using this trip data set through a time-inhomogeneous first-order Markov chain. Markov chain models are a commonly used method for uncertainty modeling, particularly in the context of EV charging loads, due to their ability to achieve high accuracy at moderate computational costs[78]. In this study, the Markov chain is employed to create trip samples between home, work, and other locations, resulting in synthetic EV driving and charging profiles. Differentiating between weekdays, weekends, and times of day, we fit a time-inhomogeneous Markov chain for our mobility simulation. We choose the Markov chain to be time-inhomogeneous, as the probability of transitioning between locations is time-dependent. When investigating the car trip dataset from the German Mobility Panel, we find that the self-reported recordings of arrival and departure times exhibit a rounding bias with ~75% of timing data points ending in a right-hand digit of either 0 or 5. These rounding biases might limit accuracy in a 1-min interval level simulation. Hence, we opt to conduct our simulation in 5-min intervals, also reducing computation time. Testing the simulation at 1-min accuracy did not affect our results.

The EV driving and charging simulation is based on an existing study adapted by separating households only according to income to reduce complexity[36]. To start our mobility simulation, we first sample the number of trips performed by each car of the lower (higher) income household on that day. In the second step, we use the first-order Markov property to define trip destinations, distances, speeds, and associated parking times depending on the start location of the current trip and its time of day (as time-inhomogeneous). Our model fits the empirical data well, with average daily driving time differing by 1.1% and an average daily trip frequency differing by 1.2% from the empirical data, respectively.

## EV and household loads

The charging logic applied does not vary by income group. However, the differing mobility behavior of lower and higher-income households impacts charging patterns. After each EV trip, the EV updates its state of charge (SOC) to reflect the distance driven. Once the EV arrives home and parks for more than 10 min, the probability of starting the charging process is determined based on the SOC as an inverse s-shaped relationship found in a 6-month German field study of 79 EV

drivers[79]. Their charging probability model defines the probability of starting the charging process as in equation (1)

$$p_{charge} = \min\left(\left(1 - \frac{1}{1 + e^{-0.15(SOC-60\%)}} c_l\right), 1\right) \quad (1)$$

for which the parameters were calibrated using an analysis of the charging behavior of EV fleets in Germany[79,80]. The factor $c_l$ can be chosen location-dependent. We focus on charging at home, representing most charging instances[81]. We adjust $c_l$ for whether a private charger is available or the charger is public and assumed to be located in front of the house. The charging process ends once the next trip is started or the EV battery is fully charged. Applying charging patterns to stop at a charge level of 80% to improve battery health would lead to similar results.

We generate the household loads via empirical sampling in two steps: First, we generate 1000 representative German household electricity load profiles for December using the Load Profile Generator[82] frequently used and validated by previous literature[83–85]. It creates representative synthetic household electricity load profiles based on a full behavior simulation of the related households[82]. We categorize these load profiles by household size. In the second step, we construct the electricity load in a typical neighborhood in rural, suburban, and urban areas for our exemplary setting of Bavaria, Germany. Therefore, we sample household load profiles via empirical sampling according to the distribution of household sizes per area type[86]. We also use area-specific distributions of households per building, as those vary by area type[87]. The respective household size and household per building distributions can be found in the Supplementary Table 1 and Supplementary Table 2.

The loads $L_{h,t}$ occurring for each house $h$ in the neighborhood at a 5 min interval time point $t \in \{1, 2, \ldots, 2016\}$ in a week are hence defined as in equation (2)

$$L_{h,t} = \sum_{hh=1}^{k}\left(e_{hh,t} + \sum_{n=1}^{n_{hh}} c_{n,t}\right) \quad (2)$$

where $k$ is the number of households in the house $h$, $e_{hh,t}$ the household electricity load profile associated with the respective household $hh$ at time $t$ and $c_{n,t}$ the charging load of an EV $n$ of the $n_{hh}$ EVs of owned by household $hh$ at time $t$.

## LV distribution grids and synthetic neighborhoods

As in earlier works, we use the SimBench low-voltage (LV) distribution grids[50,51], which are designed to represent benchmark distribution grids for Germany[52]. We opt for the SimBench grids as they allow us to analyze differing area types and the related differences in distribution grids. We perform our analysis on the SimBench LV 02 as the rural, the SimBench LV 05 as the semi-urban, and SimBench LV 06 as the urban LV grid. These encompass 95, 109, and 108 houses, respectively. To perform our analysis, we create synthetic lower (higher) income neighborhoods by allocating households sampled from the lower (higher) income data set to the SimBench grid nodes. We run a power flow analysis, and if overloads occur, we reinforce the respective overloading line or transformer.

Overloads occur, if for any grid element $g \in \{1, \ldots, G\}$ within a grid consisting of $G$ elements the related capacity $Cap_g$ is exceeded at any time point $t \in \{1, 2, \ldots, 2016\}$, meaning formulated as in equation (3)

$$\sum_{h \in H_g} L_{h,t} \le Cap_g \quad (3)$$

is violated at any time $t$ where $H_g$ is the set of all houses supplied through the grid element $g$.

Investment costs for line reinforcements in Germany are estimated as 85–125€/m[88]. We assign investment costs of 26,970€ to a 250 kVA transformer upgrade used in the rural grid and 61,730€ to a 630 kVA transformer upgrade for the suburban and urban grid[89].

## Data availability
The data used in this work is publicly available from the cited sources or can be downloaded from https://github.com/SarahASteinbach/EVgridcostInequity. Source data are provided with this paper.

## Code availability
The code used to conduct model simulations in this work is available from https://github.com/SarahASteinbach/EVgridcostInequity.

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

## Acknowledgements

We thank the Technical University of Munich for its support through the Graduate program and for providing Open Access funding. We further extend our gratitude to Prof. Dr. Gunther Friedl, Prof. Dr. Svetlana Ikonnikova, the teaching staff at TUM's Center for Energy Markets as well as School of Management for the valuable insights and comments shaping this work.

## Author contributions

S.A.S. and M.J.B. designed the study, and S.A.S. conducted the data analyses and coding. S.A.S. wrote the initial draft, and S.A.S. and M.J.B. contributed to the final manuscript.

## Funding

## Competing interests

The authors declare no competing interests.
