## [Peer Review File · Nature Communications]

Reviewers' comments:

Reviewer #1 (Remarks to the Author):

The manuscript is of significance to the emerging field of energy transitions and addresses an interesting topic of analysis. However, in its current state it contains significant flaws requiring major revision of the objectives, the literature review, the methods and data, findings, and conclusions.

I noticed that the manuscript goes from a mixed bag of topics in the first 2 ½ pages to a section 1 on results and discussion. This is odd and does not fit with best practice in publications. The paper could benefit from adding the following sections before the results and discussion section: introduction, relevant literature, methods, and data. The authors can then include their appendix with a more detailed description of methods and data. Furthermore, the manuscript could benefit from:

- Laying out in the introduction a solid discussion of why this analysis matters, how the manuscript contributes to three elements of this emerging field (EV charging, grids, and policy instruments) and what the other manuscript sections will contain.
- Then in the literature review section explaining how this manuscript compares to the established literature on the links between EV adoption-charging, grids, and policy instruments. For instance, the authors claim that income is a key determinant of EV use and charging. However, they omit relevant references noticing that multiple socioeconomic and build environment factors are associated with EV adoption (see for instance 1–4). In their own analysis, the authors also account for settlement type, a crude proxy for built environment factors. I suggest including a more systematic discussion of relevant literature. Most importantly, I suggest bringing key connections among EV adoption-charging, grids, and policy instruments to the top.
- Including a summary of the methods and data used to analyse the links and then referring to the appendix for a more detailed description. Note that I am not an expert in grid simulations, so I can't provide a qualified review of what is currently section 3.
- Including the results and discussion section that brings the results to bear on a discussion of key links among EV adoption-charging, grids, and policy instruments.
- Closing with a beefed-up conclusion section. As is the current conclusions section is quite weak.

Once these issues are addressed, the authors could revise the abstract for it to better reflect what the authors did and how their manuscript compares to existing scholarship and what its implications are for policy making.

1. Rai, V. & Robinson, S. A. Agent-based modeling of energy technology adoption: Empirical integration of social, behavioral, economic, and environmental factors. *Environ. Model. Softw.* 70, 163–177 (2015).
2. Romero Lankao, P., Wilson, A. & Zimny-Schmitt, D. Inequalities and the Future of Electric Mobility in 36 United States Cities: An Innovative Methodology and Comparative Assessment. *Energy Res. Soc. Sci.* 22 (2022).
3. Abulibdeh Ammar O., Zaidan Esmat, & Abuelgasim Abdelgadir. Urban Form and Travel Behavior as Tools to Assess Sustainable Transportation in the Greater Toronto Area. *J. Urban Plan. Dev.* 141, 05014020 (2015).
4. Ewing, R. & Cervero, R. Travel and the built environment: a meta-analysis. *J. Am. Plann. Assoc.* 76, 265–294 (2010).

Reviewer #2 (Remarks to the Author):

This paper presents analysis on household income as a critical socio-economic factor and raise a discussion on energy equity to quantify the over-proportional grid reinforcement cost impact of higher-income EV users, however the methodology is not sufficiently explained. The simulation is conducted at five-minute intervals since self-reported recordings exhibit a rounding bias with approximately 75% of timing data points ending in a right-hand digit of either 0 or 5, need justification with valid references. Here, the lack of clarity in the simulation approach raises concerns about the accuracy and validation of the simulated results. Additionally, the absence of specific details regarding the uncertainty parameters and system constraints. Based on simulated load profiles, this paper investigated the overloads occurring for below-average (lower) and above-average (higher) income rural, suburban, and urban neighborhoods, what about the intermediate (medium) layer? The paper relies heavily on assumptions and generalizations based on socio-economic factors such as income. However, the study lacks comprehensive real time data on driving patterns, EV usage, charging preferences, and income distributions across neighborhoods. The conclusions drawn from limited data may not provide better policy recommendations. While the paper claims that grid operators will have to prioritize higher-income neighborhoods due to cost asymmetries, the potential implications of this prioritization on grid stability and equitable energy distribution are not adequately explored. The paper should provide a more thorough analysis of the consequences of such prioritization and its adverse impact on different socio-economic groups. The paper must provide a more in-depth evaluation of other policy options, considering their practicality, feasibility, and impact on achieving energy equity. The paper emphasized only cost

asymmetries between higher-income and lower-income neighborhoods without considering similar disparities in other aspects of EV dynamics. A more realistic approach should be comparing with various factors influencing EV adoption and grid reinforcement costs would strengthen the research objectives.

Response to Reviewers

We would like to thank you for your valuable feedback and comments, which have allowed us to significantly improve our work. In the following, we will state the changes we introduced to reflect your guidance. Please see our comments in green font.

Reviewer #1:

The manuscript is of significance to the emerging field of energy transitions and addresses an interesting topic of analysis. However, in its current state it contains significant flaws requiring major revision of the objectives, the literature review, the methods and data, findings, and conclusions.

I noticed that the manuscript goes from a mixed bag of topics in the first 2 ½ pages to a section 1 on results and discussion. This is odd and does not fit with best practice in publications. The paper could benefit from adding the following sections before the results and discussion section: introduction, relevant literature, methods, and data. The authors can then include their appendix with a more detailed description of methods and data.

Thank you for these suggestions! We agree with your proposed structure for best practice in publications and have restructured our work to align with your advice. Following the structure requirements of Nature Communications, we had to stick to the chapter headlines Introduction (without subchapters), Results and Discussion but added subchapters in the respective sections to reflect the order suggested and better guide the reader. We separated the introduction into paragraphs of first our motivation, then the relevant literature and lastly the contributions of our work. In the results section, we first present the suggested overview of our method and the underlying data. Then we present our results and provide a detailed discussion in the closing section. The detailed description of methods and data is then available after the discussion in the appended methods.

Furthermore, the manuscript could benefit from:

- Laying out in the introduction a solid discussion of why this analysis matters, how the manuscript contributes to three elements of this emerging field (EV charging, grids, and policy instruments) and what the other manuscript sections will contain.

Thank you very much for this comment. As advised, we strengthened the motivation for our analysis in the introductions' first paragraph. We then provide a literature review and detail the contributions our work has to the different literature streams as suggested. We then close the introduction section with an outline of the other paper sections.

- Then in the literature review section explaining how this manuscript compares to the established literature on the links between EV adoption-charging, grids, and policy instruments. For instance, the authors claim that income is a key determinant of EV use and charging. However, they omit relevant references noticing that multiple socioeconomic and build environment factors are associated with EV adoption (see for instance 1–4). In their own analysis, the authors also account for settlement type, a crude proxy for built environment factors. I suggest including a more systematic discussion of relevant literature. Most importantly, I suggest bringing key connections among EV adoption-charging, grids, and policy instruments to the top.

Thank you very much for your comment and for sharing your paper recommendations that we included in our literature review. We follow your suggestions and structure the different literature streams in the literature review part within the introduction section (starting in paragraph 2) accordingly. Each of the following literature streams discusses the most important literature within an individual paragraph:

- Paragraph 2) A detailed literature review about the general problem of simultaneous charging and pressure on the grids
- Paragraph 3) A detailed literature review on the various socioeconomic and environment factors associated with EV adoption including the papers suggested by the reviewer.
- Paragraph 4) A detailed literature review about car usage and the socio-economic differences in driving pattern.
- Paragraph 5) A discussion on the general aspects of fairness with corresponding literature brought to the energy context including the current policy and regulatory status.

- Including a summary of the methods and data used to analyse the links and then referring to the appendix for a more detailed description. Note that I am not an expert in grid simulations, so I can't provide a qualified review of what is currently section 3.

Thank you very much for this recommendation according to which we have restructured our work! As our previous manuscript was forwarded from Nature Energy, we had to follow their guidelines to have all methods in the appendix. Now, we have included a summary of the methods and data used in the main part of our article, referring to the appendix only for more details. The most important parts with

an overview of our model, scenario analyzed and driving patterns as well as the load profile implications are now available in the main text in Section 2.1-2.3 to give the reader a better understanding of our approach before presenting the results.

- Including the results and discussion section that brings the results to bear on a discussion of key links among EV adoption-charging, grids, and policy instruments.

Thank you for this suggestion! We fully agree. We have now included the results and discussion section after outlining the methods and data used. We now provide a separate subsection for the policy options (2.7) and have expanded on their discussion.

- Closing with a beefed-up conclusion section. As is the current conclusions section is quite weak.

Thank you for this recommendation! Following your guidance, we rewrote the final discussion chapter to better emphasize the impact of these cost imbalances and further strengthened the policy discussion within the section.

Once these issues are addressed, the authors could revise the abstract for it to better reflect what the authors did and how their manuscript compares to existing scholarship and what its implications are for policy making.

We agree and have adapted the abstract to better reflect our approach and state the policy implications. Furthermore, we give the reader a short statement to the contribution compared to existing literature.

1. Rai, V. & Robinson, S. A. Agent-based modeling of energy technology adoption: Empirical integration of social, behavioral, economic, and environmental factors. *Environ. Model. Softw.* 70, 163–177 (2015).
2. Romero Lankao, P., Wilson, A. & Zimny-Schmitt, D. Inequalities and the Future of Electric Mobility in 36 United States Cities: An Innovative Methodology and Comparative Assessment. *Energy Res. Soc. Sci.* 22 (2022).
3. Abulibdeh Ammar O., Zaidan Esmat, & Abuelgasim Abdelgadir. Urban Form and Travel Behavior as Tools to Assess Sustainable Transportation in the Greater Toronto Area. *J. Urban Plan. Dev.* 141, 05014020 (2015).
4. Ewing, R. & Cervero, R. Travel and the built environment: a meta-analysis. *J. Am. Plann. Assoc.* 76, 265–294 (2010).

Thank you very much for these recommendations. We included them in the literature review.

Thank you very much for all your valuable advice!

Reviewer #2:

This paper presents analysis on household income as a critical socio-economic factor and raise a discussion on energy equity to quantify the over-proportional grid reinforcement cost impact of higher-income EV users, however the methodology is not sufficiently explained.

Thank you very much for your advice. We follow your recommendations and have therefore included more detailed explanations of our methodology in Section 2.1-2.3 & 4 and have added in additional details to guide the reader through our approach as described in the following.

The simulation is conducted at five-minute intervals since self-reported recordings exhibit a rounding bias with approximately 75% of timing data points ending in a right-hand digit of either 0 or 5, need justification with valid references. Here, the lack of clarity in the simulation approach raises concerns about the accuracy and validation of the simulated results.

Thanks a lot for making us aware of this potential for concern! We have expanded on this sentence to improve clarity of our reasoning for the reader: “When investigating the car trip dataset from the German Mobility Panel, we find that the self-reported recordings of arrival and departure times exhibit a rounding bias with approximately 75% of timing data points ending in a right-hand digit of either 0 or 5. These rounding biases might limit accuracy in a one minute interval level simulation. We hence opt to conduct our simulation in five-minute intervals, also reducing computation time. Testing the simulation at one minute accuracy did not affect our results.”

By that, we aim to avoid confusion for the reader since the rounding biases actually come from the mobility data set itself. For example, someone who actually left the house at 3:01 pm still reported that he left at 3:00pm. We also did perform testing on a 1 min interval level, however, the results remained unchanged. Further, the aforementioned rounding biases within the mobility data set limit the accuracy in a 1 min interval level simulation, creating artificial peaks in arrival and departure times every 5 minutes. We therefore opted for a 5 min interval level, also to reduce computation time.

Additionally, the absence of specific details regarding the uncertainty parameters and system constraints.

Thank you for enabling us to clarify this aspect and better guide the reader through our model. We now have included clarifications in Section 2.1 (most relevant), 4.1 and 4.2.

Section 2.1 now includes: “For both neighborhood types,

1. we populate the grids with the related income group (above- or below-average-income households) by sampling from the household distributions of each neighborhood and grid type.
2. we assign the related EV adoption level and model choice by randomly allocating EVs depending on the income group's EV adoption and model choice distribution.
3. we model the driving patterns for each EV depending on the income group by a time-homogeneous Markov chain simulation.
4. we derive the EV charging loads resulting from the EV driving patterns based on the charging probability as well as future car trips planned.
5. we empirically sample representative household electricity load profiles on a household level.
6. we consolidate the EV charging loads and household electricity load profiles by adding up all loads on a house-level.
7. we perform a power flow analysis, and if overloads occur, we reinforce the respective overloading grid element.
8. we calculate grid reinforcement costs to resolve these overloads for each grid and neighborhood type.”

The clarifications now included in Section 2.1, 4.1 and 4.2 describe the uncertainty parameters and system constraints in more detail:

We hereby clarify for the reader that the uncertainty is introduced as follows: EV ownership is randomly assigned based on the respective probability to own a car as well as the probability of the car being an EV. EV models are randomly assigned based on the respective model distribution. For the EV driving patterns, car trip data is simulated as a time-inhomogeneous Markov Chain for each EV (Section 2.1 and 4.1). Based on the state of charge, location of the EV and parking times, a charging probability is calculated that then is used for the binary decision to initiate the charging process (Section 4.2).

For the household loads, uncertainty is introduced in the model via empirical sampling of representative electricity load profiles based on the sampled household sizes and households per building in accordance to their respective distributions in the corresponding neighborhood (Section 4.2).

The system constraints include the underlying assumptions for household sizes, households per building, EV adoption scenario, EV model distribution as well as grid constraints of the representative grid models used (Section 2.2, 4.1-4.4).

Based on simulated load profiles, this paper investigated the overloads occurring for below-average (lower) and above-average (higher) income rural, suburban, and urban neighborhoods, what about the intermediate (medium) layer?

We appreciate your thoughts on this and fully agree! We initially considered separating households into narrower income groups. In our current separation, we opt for a 50:50 split into above-average and below-average income households. As we already see very significant effects for this separation and data availability for narrower income groups is generally sparse, we opted to explore only this separation. We have included the following remark in the main part of the article, to make our reasoning more clear to the reader:

“The differences observed between above- and below-average neighborhoods with a 50:50 split already showed such significant cost imbalances that a further split of the population in smaller groups would most likely only lead to even more extreme results. Due to smaller samples and less data availability, we, however, would not receive stronger or more robust findings.”

The paper relies heavily on assumptions and generalizations based on socio-economic factors such as income. However, the study lacks comprehensive real time data on driving patterns, EV usage, charging preferences, and income distributions across neighborhoods. The conclusions drawn from limited data may not provide better policy recommendations.

Thank you for your advice! To give the reader a better understanding of the origin of our data used, we have now included remarks on our model leveraging real-life data in the abstract, Section 2.1 as well as Section 4.1. Furthermore, we have included a summary of the methods and data used in the main part of our article to allow for a better overview of our approach, referring to the appendix for more detail.

For our driving and car usage patterns across neighborhoods and income groups, we are using representative, real-life mobility behavior data from the German Mobility Panel (<https://mobilitaetspanel.ifv.kit.edu/index.php>), which is a 30 year project of a renowned German university collecting mobility data for households. The underlying driving data has an accuracy of 1 minute on all tracked trips created in 2019-2020 as a representative driving survey. This data also allows for separation by income groups as derived from Statistisches Bundesamt [63,64], which collects statistical data on behalf of the German government. To assign car models, we use current data from Kraftfahrt Bundesamt, which tracks all national car registrations (https://www.kba.de/DE/Statistik/Produktkatalog/produkte/Fahrzeuge/fz11/fz11_gentab.html). To account for real-life charging preferences, we leverage the results from two German charging behavior studies [73,74].

While the paper claims that grid operators will have to prioritize higher-income neighborhoods due to cost asymmetries, the potential implications of this

prioritization on grid stability and equitable energy distribution are not adequately explored. The paper should provide a more thorough analysis of the consequences of such prioritization and its adverse impact on different socio-economic groups.

Thank you very much for this valuable and very interesting thought! We have now linked the aspect of prioritization to grid stability and energy equity in Section 2.6 as follows: “This effect could be aggravated further, if grid operators decide to prioritize higher income neighborhoods in their grid planning, potentially making some households pay higher electricity prices long before they can profit from more stable grids in their neighborhoods. Furthermore, this prioritization could lead to energy access equity issues between neighborhoods, as for example already observed for the case of solar PVs in the US by [9].”

The paper must provide a more in-depth evaluation of other policy options, considering their practicality, feasibility, and impact on achieving energy equity.

Thank you very much for this advice! We followed your suggestion and have now included a new subsection of policy options (2.7) and have expanded on their discussion.

The paper emphasized only cost asymmetries between higher-income and lower-income neighborhoods without considering similar disparities in other aspects of EV dynamics. A more realistic approach should be comparing with various factors influencing EV adoption and grid reinforcement costs would strengthen the research objectives.

Thanks a lot for this recommendation! We agree that including various factors influencing EV charging would further strengthen the analysis. However, limited data availability did not allow for the inclusion of other factors with an appropriate accuracy and resource limitation. We therefore focussed on income as a factor which leads to strong cost implications and is especially critical from an energy equity perspective.

Thank you very much for all your helpful recommendations!

REVIEWERS' COMMENTS

Reviewer #2 (Remarks to the Author):

I have no more comments. I see the revision is well done